# Evaluation and comparison of scoring systems for predicting stone-free status after flexible ureteroscopy for renal and ureteral stones

François Richard[1,2], Jonathan Marguin[1,2], Alexandre Frontczak[1,2], Johann Barkatz[1,2], Loic Balssa[1,2], Stéphane Bernardini[1,2], Eric Chabannes[1,2], Guillaume Guichard[1,2], Hugues Bittard[1,2], François Kleinclauss[1,2,3]*

1 Department of Urology and Renal Transplantation, University Hospital of Besancon, Besancon, France, 2 Université de Franche-Comté, Besançon, France, 3 "Nanomedicine Lab, Imagery and Therapeutics", EA 4662, Besançon, France

* francois.kleinclauss@univ-fcomte.fr

## Abstract

### Objective

To evaluate four predictive scores for stone-free rate (SFR) after flexible ureterorenoscopy (f-URS) with holmium-YAG laser fragmentation of renal and ureteral lithiasis.

### Methods

We carried out a retrospective analysis of 800 f-URS procedures performed in our institution between January 2009 and December 2016. For each procedure, a single surgeon calculated the following scores: S.T.O.N.E score; Resorlu Unsal Stone Score (RUSS); modified Seoul National University Renal Complexity (S-ReSC) score; and Ito's score.

### Results

Overall SFR was 74.1%. Univariate analysis demonstrated that stone size (p<0.0001), stone volume (p<0.0001), stone number (p = 0.004), narrow lower pole infundibulopelvic angle (IPA) (p = 0.003) and lower pole location + IPA <45° (p = 0.011) were significantly associated with SFR. All scores differed between the stone-free and non-stone-free groups. Area under the curve of the receiving operator characteristics curve was calculated for each score: 0.617 [95%CI: 0.575–0.660] for the S.T.O.N.E score; 0.644 [95%CI: 0.609–0.680] for the RUSS; 0.651 [95%CI: 0.606–0.697] for the S-ReSC score; and 0.735 [95%CI: 0.692–0.777] for Ito's nomogram.

### Conclusion

All four scores were predictive of SFR after f-URS. Ito's score was the most sensitive. However, the performance of all scores in this analysis was lower than in developmental studies.

**Data Availability Statement:** All relevant data are within the paper and its Supporting Information files.

**Funding:** The author(s) received no specific funding for this work.

**Competing interests:** The authors have declared that no competing interests exist.

## Introduction

The prevalence and incidence of renal and ureteral lithiasis are increasing worldwide [1, 2]. Surgical treatments are evolving and flexible ureterorenoscopy (f-URS) has been shown to outperform extracorporeal shockwave lithotripsy (ESWL) and percutaneous nephrolithotomy (PCNL) in a number of indications [3, 4]. This depends mainly on the size, number and location of the stones. ESWL is a minimally invasive technique but often requires several procedures. PCNL seems to be the most successful but is the most invasive procedure [5]. f-URS also gives excellent results and is associated with a low complication rate [6]. Furthermore, f-URS is a safe and efficient option in specific populations such as obese patients [7].

Many studies have evaluated the factors affecting the success rate of these procedures. Preoperative scores and nomograms have been developed and are available to predict the stone-free rate (SFR) with varying degrees of accuracy. Several factors affecting the outcome of ESWL and PCNL have been described and nomograms have been established [8–11].

For f-URS, four scores have been developed. Two of these have been compared and validated in different cohorts: the Resolu Unsal Stone Score (RUSS) and modified Seoul National University Renal Stone Complexity (S-ReSC) score [12, 13]. To our knowledge, the other two scores, the S.T.O.N.E score described by Molina et al. [14] and Ito's nomogram [15] have not been validated in any other cohorts.

Our aim was to evaluate these four scores in a large group of patients with renal and ureteral stones treated by f-URS and holmium fragmentation.

## Materials and methods

### Study population

This large retrospective study evaluated all f-URS procedures for upper urinary tract calculi performed in our institution between January 2009 and December 2016. A total of 1011 consecutive procedures were recorded: 211 were excluded from the study due to lack of preoperative imaging and insufficient post-operative follow-up. Thus 800 procedures were analysed. The S.T.O.N.E score analysis was performed on the entire cohort ('ureteral and renal' group (UR): 800 procedures). The other three scores were developed for the renal stone only. They were analysed in the subgroup of patients with renal stones only ('renal only' group (RO): 669 procedures).

### Surgical procedures

f-URS was performed by several experienced surgeons following a single standardized protocol. All details of the surgical procedure have been described previously [16]. Briefly, all procedures were performed under general anaesthesia. Patients were installed in the lithotomy position (with the leg on the treated side in semi-extension to straighten the ureter). The procedure began with rigid cystoscopy to insert a stiff guidewire up to the kidney. Ureteral dilation and an access sheath were used when necessary (mainly 12F/14F, Cook Medical®, Bloomington, IN, USA; or 11/13 F, Boston Scientific®, Natick, MA, USA) and were placed under radioscopic guidance. An irrigation pressurizing system was used routinely. Flexible ureteroscopes were Flex-X2TM, Storz®; Tuttlingen, Germany, or Olympus URF-V; Olympus®, Tokyo, Japan. Stones were fragmented with a holmium:YAG laser (272 or 365l m fibres). Once fragmentation was completed, the fragments were extracted with different nitinol stone baskets. After stone removal, a double pigtail indwelling ureteral stent (7F) was placed at the surgeon's discretion.

## Preoperative and postoperative evaluation

The following data were recorded: epidemiological characteristics (age, gender, body mass index (BMI), medical treatment, medical history), characteristics of the stones (localization, size, density, composition), surgical treatment (duration of the procedure, ureteral dilation, ureteral access sheath (UAS)), surgical and anaesthesia complications according to the Clavien-Dindo classification [17]. All patients included were evaluated by performing a preoperative computed tomography (CT) scan.

## Score analysis

**S.T.O.N.E. score.** The S.T.O.N.E score was calculated using five parameters: S: size of stone; T: topography or location; O: obstruction; N: number of stone(s); E: evaluation of Hounsfield Units (HU). Each variable is graded from 1 to 3 and the grades are added to give a final score (5 to 15 points). A high total score is predictive of a failure f-URS outcome.

**RUSS score.** The RUSS score considers four parameters (1 point for each of the four criteria): stone size >20 mm; lower pole location with infundibulopelvic angle (IPA) <45˚; stone number in different calyces >1; abnormal renal anatomy; with a total score ranging from 0 to 4.

**Modified ReSC.** This score depends on two variables: stone number and stone location. Nine locations are identified and one point is given for each location involved, except for the inferior pole which is allocated 2 points.

**Ito's nomogram.** This score is calculated from five variables: stone volume (0/5/8/13 point(s)); presence of lower pole calculi (0/5 point(s)); operator with experience of >50 f-URS (0/3 point(s)); stone number (0/2 point(s)); presence of hydronephrosis (0/2 point(s)). Total nomogram score (0 to 25 points) is derived from adding up the individual scores. In contrast to the other three scores, a high total score is predictive of a successful f-URS outcome.

SFR was defined by the total absence of residual stone after rigorous examination at the end of the procedure. After stone removal, all calyces were inspected on a visual display under radioscopic control. If fragments or dust remained, scans were repeated up to 6 months to search for new aggregates. The patient was systematically considered to be non-stone-free (SF) if imaging was positive for the presence of stones. If postsurgical inspection was totally clear, the absence of clinical complications or CT proof of stone recurrence during the 6-month follow-up confirmed that the patient was SF.

## Ethical approval

This study was approved by the Institutional Ethics Committee of University Hospital of Besancon. All data were anonymized. The Ethics committee waived the requirement for informed consent.

## Statistical analysis

Analysis was performed using the Chi-square test for categorical variables and Mann Whitney U test for continuous variable. A p value of <0.05 was considered statistically significant. Area under the curve of receiver operating characteristics curve (AUC-ROC) was used to evaluate the performance of the scores. All statistical analyses were performed using XLSTAT Premium software version 20.1 (Addinsoft®, Brooklyn, NY, USA).

# Results

## Patient characteristics and surgical outcome

A total of 800 procedures carried out between January 2009 and December 2016 were analysed. The characteristics of the patients and procedures are shown in Table 1.

Overall, there were 500 males (62.5%) and 300 females (37.5%), mean (SD) age was 52.3 ± 16.8 years and mean BMI was 26.6 ± 5.9 kg/m$^2$. Mean diameter of the stones was

**Table 1. Patient demographics, stone characteristics, procedure characteristics and complications.**

| | Stone group | |
|---|---|---|
| | Ureteral and renal (UR) | Renal only (RO) |
| **Patients** | | |
| Number of procedures | 800 | 669 |
| Number of patients | 577 | 484 |
| Mean age (years) | 52.3 ± 16.8 | 52.5 ±16.7 |
| Gender (n, %) | | |
| Female | 300 (37.5%) | 264 (39.5%) |
| Male | 500 (62.5%) | 405 (61.5%) |
| Mean BMI (kg/m$^2$) | 26.6 ± 5.9 | 26.6± 6.0 |
| **Stones** | | |
| Size (mm) | 10.9 ± 8.8 | 11.5 ± 9.0 |
| Procedures with stone >20 mm (n, %) | 84 (10.5%) | 87 (13.0%) |
| Average Hounsfield Unit | 780.2 ± 334.1 | 783.0 ± 337.5 |
| Mean number of stones | 1.9 ± 1.3 | 2.0 ± 1.4 |
| **Procedure** | | |
| Previous treatment (n, %) | | |
| RIRS | 146 (18.3%) | 138 (20.6%) |
| ESWL | 89 (11.1%) | 79 (11.8%) |
| PCNL | 28 (3.5%) | 28 (4.2%) |
| Pre-stenting (n, %) | | |
| Double J stent | 515 (64.4%) | 426 (63.7%) |
| Percutaneous nephrostomy | 6 (0.8%) | 2 (0.3%) |
| Duration (min) | 93.1 ± 41.7 | 97.1 ± 42.5 |
| Ureteral dilation (n, %) | 40 (5.0%) | 35 (5.2%) |
| Access sheath (n, %) | 698 (87.3%) | 606 (90.6%) |
| Hospital stay (days) | 3.3 ± 2.0 | 3.3 ± 1.9 |
| Post-operative stenting (n, %) | | |
| None | 108 (13.5%) | 78 (11.7%) |
| Mono J Stent | 124 (15.5%) | 104 (15.6%) |
| Double J stent | 568 (71.0%) | 489 (73.1%) |
| SFR | 593 (74.1%) | 482 (72.1%) |
| **Complications (n, %)** | | |
| All grades | 85 (10.6%) | 69 (10.3%) |
| Clavien I | 40 (5.0%) | 32 (4.8%) |
| Clavien II | 41 (5.1%) | 33 (4.9%) |
| Clavien III | 3 (0.4%) | 3 (0.5%) |
| Clavien IV | 0 | 0 |
| Clavien V | 1 (0.1%) | 1 (0.2%) |

BMI: body mass index; RIRS: retrograde intra renal surgery; ESWL: extracorporeal shockwave lithotripsy; PCNL: percutaneous nephrolithotomy; SFR: stone-free rate.

**Table 2. Univariate analysis of patient and stone characteristics.**

| | | Stone-free | | p value |
|---|---|---|---|---|
| | | Yes | No | |
| UR procedure (n, %) | UR | 593 (74.1%) | 207 (25.9%) | |
| RO procedure (n, %) | RO | 482 (72.1%) | 187 (27.9%) | |
| Age (years) | UR | 52.0 ± 16.7 | 53.2 ± 17.1 | 0.405 |
| BMI (kg/m$^2$) | UR | 26.4 ± 5.7 | 27.1 ± 6.2 | 0.125 |
| Size (mm) | UR | 9.2 ± 5.8 | 16.6 ± 11.8 | <0.0001 |
| Volume (mm$^3$) | UR | 1348.8 ± 10634.0 | 7924.0 ± 27335.0 | <0.0001 |
| No. of stones | UR | 1.8 ± 1.3 | 2.1 ± 1.4 | 0.004 |
| | RO | 2.0 ± 1.4 | 2.2 ± 1.4 | 0.046 |
| Stone density (HU) | UR | 788.6 ± 342.3 | 756.2 ± 308.9 | 0.323 |
| Pre-stenting (yes/total and % yes) | UR | 385/593 | 130/207 | 0.564 |
| | | (65.0%) | (62.8%) | |
| Lower pole location (yes/total and % yes) | RO | 308/482 | 128/187 | 0.268 |
| | | (63.9%) | (68.4%) | |
| IPA in degrees | RO | 51.5 ± 9.1 | 48.6 ± 9.1 | 0.003 |
| Lower pole location + IPA <45˚ (yes/total and % yes) | RO | 71/482 | 43/187 | 0.011 |
| | | (14.7%) | (23.0%) | |

HU: Hounsfield Unit; UR: ureteral and renal; RO: renal only; IPA: infundibulopelvic angle.

10.9 ± 8.8 mm, mean renal stone density was 780.2 ± 334.1 HU and mean number of stones per patient was 1.9 ± 1.3. Repartition of chemical composition: 66% were calcium oxalate, 14% were calcium phosphate, 9% were uric acid and 11% were other. There were 515 (64.4%) double J stents. Ureteral dilation was required in 40 cases (5%), an access sheath was used in 698 procedures (87.3%) and mean operation time was 93.1 ± 41.7 min. SFR was 74.1% (n = 593). The complication rate was 10.6% (n = 85) and most complications were mild: Clavien-Dindo grade III complications were observed in three patients only (0.4%). After exclusion of ureteral procedures (131 patients), the RO group had similar characteristics, with a SFR of 72.1%.

## Analysis of patient and stone characteristics

The characteristics of the patients and stones were compared between the SF and non-SF groups (Table 2).

Univariate analysis showed that stone size (9.2 ± 5.8 vs. 16.6 ± 11.8 mm, p<0.0001), stone volume (1348.8 ± 10634.0 vs. 7924.0 ± 27335.0 mm$^3$, p<0.0001), stone number (1.8 ± 1.3 vs. 2.1 ± 1.4. p = 0.004), lower pole IPA (51.5 ± 9.1˚ vs. 48.6 ± 9.1˚, p = 0.003) and lower pole location + IPA <45˚ (14.7% vs. 23.0%, p = 0.011) were significantly different between the two groups. No statistical difference was found for stone density (788.6 ± 342.3 vs. 756.2 ± 308.9 HU, p = 0,323), pre-stenting (65.0% vs. 62.8%, p = 0.564) and lower pole location (63.9% vs. 68.4%, p = 0.268).

## Score analysis

By univariate analysis all scores differed between the SF and non-SF groups (Table 3): 9.223 ± 1.778 vs. 10.005 ± 1.685 (p<0.0001) for the S.T.O.N.E score; 0.064 ± 0.319 vs. 0.471 ± 0.778 (p<0.0001) for the RUSS score; 2.643 ± 1.580 vs. 3.455 ± 1.717 (p<0.0001) for the Modified ReSC score; and 14.05 ± 5.24 vs. 9.17 ± 5.82 (p<0.0001) for Ito's nomogram.

AUC ROC was calculated for the four scores (Fig 1).

**Table 3. Univariate analysis of scores and nomogram.**

| | Stone-free | | p value |
|---|---|---|---|
| | **Yes** | **No** | |
| **S.T.O.N.E score (UR: 800 procedures)** | 9.223 ± 1.778 | 10.005 ± 1.685 | <0.0001 |
| S (size) (1–3 pts) | 2.150 ± 0.598 | 2.623 ± 0.560 | <0.0001 |
| T (topography) (1–3 pts) | 2.364 ± 0.766 | 2.551 ± 0.636 | 0.002 |
| O (obstruction) (1–3 pts) | 1.374 ± 0.484 | 1.406 ± 0.492 | 0.424 |
| N (number of stones) (1–3 pts) | 1.644 ± 0.805 | 1.831 ± 0.835 | 0.005 |
| E (evaluation of HU) (1–3 pts) | 1.696 ± 0.805 | 1.623 ± 0.784 | 0.252 |
| **RUSS score (RO: 669 procedures)** | 0.064 ± 0.319 | 0.471 ± 0.778 | <0.0001 |
| Stone size >20 mm (1pt per 10 mm) | 20/482 | 59/187 | <0.0001 |
| (yes/total and % yes) | (4.1%) | (31.6%) | |
| Lower pole location + IPA <45˚ (1 pt) | 71/482 | 43/187 | 0.011 |
| (yes/total and % yes) | (14.7%) | (23.0%) | |
| Stone number in different calyces >1 (1 pt) | 110/482 | 70/187 | 0.0001 |
| (yes/total and % yes) | (22.8%) | (37.4%) | |
| Abnormal renal anatomy (1 pt) | 2/482 | 2/187 | 0.324 |
| (yes/total and % yes) | (0.4%) | (1.1%) | |
| **Seoul modified (RO: 669 procedures)** | 2.643 ± 1.580 | 3.455 ± 1.717 | <0.0001 |
| **Ito's score (RO: 669 procedures)** | 14.05 ± 5.24 | 9.17 ± 5.82 | <0.0001 |
| Stone volume (n, %) | | | <0.0001 |
| ≤ 500 (13 pts) | 303 (62.9%) | 45 (24.1%) | |
| 500 <v≤ 1000 (8 pts) | 85 (17.6%) | 39 (20.9%) | |
| 1000 <v≤ 2000 (5 pts) | 50 (10.4%) | 29 (15.5%) | |
| >2000 (0 pts) | 44 (9.1%) | 74 (39.6%) | |
| No lower pole calculi (yes/total) | 174/482 | 59/187 | 0.268 |
| (%yes) (5 pts) | (36.1%) | (31.6%) | |
| Operator experience ≥ 50 (yes/total) | 244/482 | 97/187 | 0.772 |
| (%yes) (3 pts) | (50.6%) | (51.8%) | |
| Absence of hydronephrosis (yes/total) | 294/482 | 113/187 | 0.893 |
| (% absent) (2 pts) | (61.0%) | (60.4%) | |
| Number of stones (solitary/total) | 223/482 | 73/187 | 0.09 |
| (% solitary) (2 pts) | (46.3%) | (39.0%) | |

UR: ureteral and renal; RO: renal only; RUSS: Resolu-Unsal Stone Score; HU: Hounsfield Unit; IPA: infundibulopelvic angle.

## Discussion

The indications for f-URS have increased considerably over the past few years due to improvements in materials and techniques. f-URS is now the preferred technique before ESWL and PCNL and is associated with few complications, which are mainly low grade [18]. However, prediction of the results of f-URS using scores is essential for patient management.

In our patient cohort, scores were significantly different between the SF and non-SF groups. The AUC ROC for the four scores were predictive of SFR at an intermediate level of prediction [19]. The AUC ROC for the scores were lower than in developmental studies: Jung et al. 0.766 for the ReSC score [13], Molina et al. 0.764 for the S.T.O.N.E score [14], Resorlu et al. [12] did not calculate the AUC ROC for the RUSS score, and Ito et al. reported a higher AUC ROC: 0.87 [15]. However, our lower AUC ROC concurs with two other validations: Erbin et al. reported an AUC ROC of 0.655 for the RUSS and 0.593 for the modified ReSC [20]. Park et al.

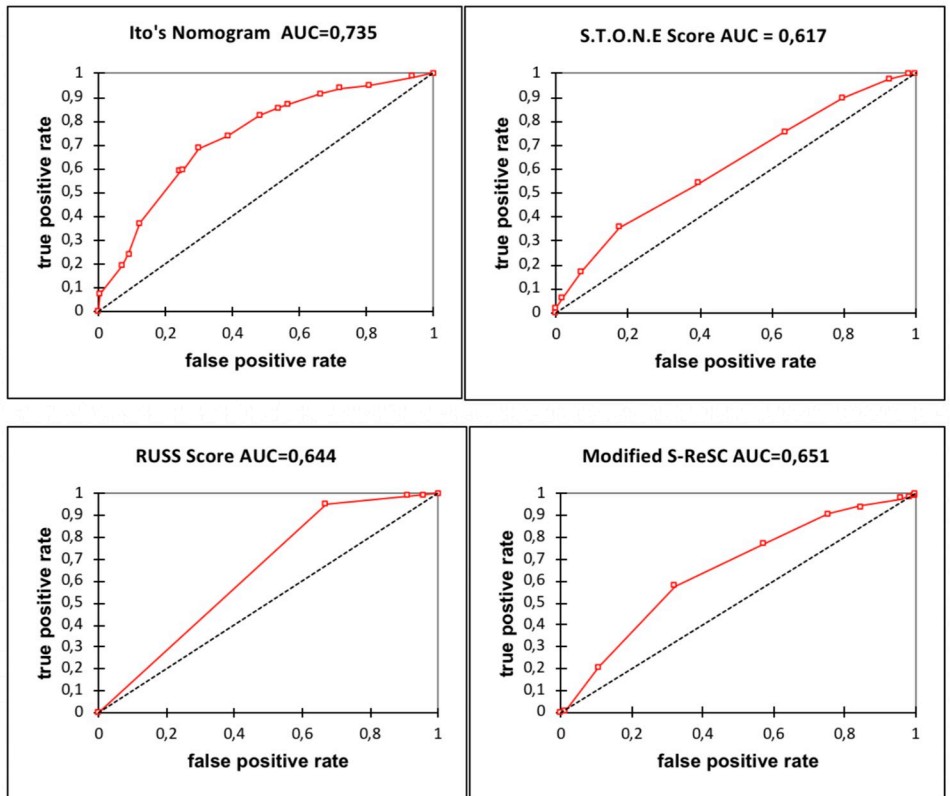

**Fig 1. ROC curves for Ito's nomogram, S.T.O.N.E score, RUSS score and modified S- ReSC score.** S.T.O.N.E score AUC ROC was 0.617 [95%CI: 0.575–0.660], RUSS AUC ROC was 0.644 [95%CI: 0.609–0.680], modified S-ReSC AUC ROC was 0.651 [96%CI: 0.606–0.697] and Ito's nomogram AUC ROC was 0.735 [95%CI: 0.692–0.777].

reported an AUC ROC of 0.701 for the modified ResC [21]. Furthermore, external validation of the PCNL score also gave an inferior AUC ROC to developmental studies [22, 23].

Although the differences between the f-URS scores were not significant, the nomogram of Ito et al. [15] appears to be the most predictive, in accordance with previous studies. If we consider each variable individually, obstruction was not considered a significant factor in our study. This supports the results of Molina et al. who failed to demonstrate the significance of obstruction (0.5 [95%CI: -1.1–0.1], p = 0.077) [14]. The impact of obstruction is difficult to evaluate as the rate of pre-stenting was high among our patients. Furthermore, Rubenstein et al. demonstrated a difference in the presence of pre-stenting in contrast to a previous study in our centre [24, 25]. No significant difference has been reported for density, as reported by Molina et al. [14]. These two variables (obstruction and density) have a tendency to lower the predictive value of the S.T.O.N.E score. Lower pole location and IPA angle is a unique paring. Our study confirms the findings of Resorlu et al., although calculating the IPA angle appears to give slightly superior results. IPA was demonstrated to be significant by Inoue et al. and Jessen et al., even using a cut-off of 30˚ [26, 27]. Measurements were made as described by Elbahnasy et al., but Sampaio et al. suggested another method using stone position [28, 29]. It is also the only score that considers musculoskeletal deformities and renal malformations. To date, no study has assessed the effect of renal malformation on SFR after f-URS. Resorlu et al. found a SFR of 66.7% (10/15) vs. 87.5% (168/192) (p = 0.04), respectively in patients with or without renal malformations. Although renal malformation should intuitively lower the SFR, the effect

seems to be moderate and needs to be confirmed. Our study did not identify enough cases of renal malformation to draw any conclusions. The four criteria of the score are also very restrictive, giving excellent specificity but poor sensitivity. The modified ReSC score focusses on two variables: stone number and location. It is the easiest and most practical score to use. However, visualisation of the calyceal is not always easy. The score described by Ito et al. [15] appears to be the most predictive. However, the use of a paper nomogram is required. Weighting of these variables, particularly 'stone volume', seems to enhance its results ('stone volume' graded from 0 to 13 points, on a total score ranging from 0 to 25 points). In contrast to the report of Ito et al., stone volume in our study was estimated using only maximal diameter. The performance of this variable did not seem to be affected by this quicker method. The other variables did not appear to be significant in our study. As in a previous study in our centre, stone location, in particular lower pole location, did not have any impact on the efficacy and morbidity of f-URS [30].

Our study has several advantages and limitations. In addition to our cohort being the largest described to date, the study also involved a number of surgeons (12 operators) with different levels of experience (residents, novices and experienced surgeons) reflecting real-life urological practice. Furthermore, only one person was involved in calculating the four scores. The limitations of our study include its retrospective nature, a source of important bias. A SFR of 74% is similar to the external validation studies of Erbin et al. and Parks et al. (respectively 70.1% and 73%). For statistical and practical reasons, the definition of SFR is a single stone for all scores. This is precise and is defined by a total absence of stone and not by an absence of stone sized >4 or >2 mm. This definition can contribute to decrease the AUC ROC. One of the limitations of our study is the absence of systematic post-operative imaging. However, endoscopic and fluoroscopic end of procedure inspection was always performed. This technique has shown good sensitivity and specificity at detecting residual stone fragments [31]. The simultaneous evaluation of these four scores on a single cohort requires non-restrictive inclusion criteria, whereas some score variables were exclusion criteria in other studies. Our choice better reflects daily practice.

A large, multicentre, prospective study, recording all variables, would allow the validity of these scores to be defined. An ideal score should be sensitive and available for all patients, but it will be difficult to combine performance and simplicity/efficiency. Based on our results it is difficult to rank the different scores according their accuracy. We found that four main factors: stone size, stone number, lower pole and IPA <45˚ affected the scores. Moreover, we should also ask what the clinical impact of these scores is. Independent of the information given to patients and possibly to help in the choice of surgical technique, these kind of scores are not clinically useful and are not used routinely due to their difficulty and the time required to calculate them. The RUSS score needs IPA angle definition on a computed tomography scan and the Ito score requires the use of a nomogram and measurement of the stone volume. Conversely, the S.T.O.N.E. score can be calculated mentally in daily practice and the Seoul score would be easiest to use.

Future studies should also analyse inter- and intra-observer variability. Automatic analysis of pre-operative images by artificial intelligence programs could occur in the near future.

## Conclusion

A number of scores have been established to predict SFR after RIRS. The four scores evaluated in this study were all predictive of SFR after f-URS. The score of Ito et al. appeared to be the most sensitive. We also tested the ability of the score to predict complications. None of the

four scores were predictive of the complication rate in RIRS. Other nomograms could be developed in this area.

## Supporting information

**S1 Dataset.**
(XLSX)

## Author Contributions

**Conceptualization:** François Kleinclauss.

**Data curation:** François Richard, Jonathan Marguin, François Kleinclauss.

**Formal analysis:** François Richard, François Kleinclauss.

**Investigation:** François Richard, Jonathan Marguin, François Kleinclauss.

**Methodology:** François Richard, François Kleinclauss.

**Project administration:** François Richard, François Kleinclauss.

**Resources:** François Richard, François Kleinclauss.

**Software:** François Richard, François Kleinclauss.

**Supervision:** François Richard, Alexandre Frontczak, Johann Barkatz, Loic Balssa, Stéphane Bernardini, Eric Chabannes, Guillaume Guichard, Hugues Bittard, François Kleinclauss.

**Validation:** François Richard, François Kleinclauss.

**Visualization:** François Richard, François Kleinclauss.

**Writing – original draft:** François Richard, Eric Chabannes, François Kleinclauss.

**Writing – review & editing:** François Richard, Eric Chabannes, François Kleinclauss.

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
