## [Decision Letter · Decision Letter 0]

15 Apr 2020

PONE-D-20-00720

Evaluation and comparison of different nomograms for predicting stone-free status after flexible ureteroscopy for renal and ureteral stones

PLOS ONE

Dear Mr. Kleinclauss,

Thank you for submitting your manuscript to PLOS ONE. After careful consideration, we feel that it has merit but does not fully meet PLOS ONE’s publication criteria as it currently stands. Therefore, we invite you to submit a revised version of the manuscript that addresses the points raised during the review process.

We would appreciate receiving your revised manuscript by May 30 2020 11:59PM. To enhance the reproducibility of your results, we recommend that if applicable you deposit your laboratory protocols in protocols.io, where a protocol can be assigned its own identifier (DOI) such that it can be cited independently in the future. For instructions see: http://journals.plos.org/plosone/s/submission-guidelines#loc-laboratory-protocols

We look forward to receiving your revised manuscript.

Kind regards,

Antonio Riveiro Rodríguez, PhD

Academic Editor

PLOS ONE

2. In ethics statement in the manuscript and in the online submission form, please provide additional information about the patient records/samples used in your retrospective study. Specifically, please ensure that you have discussed whether all data/samples were fully anonymized before you accessed them and/or whether the IRB or ethics committee waived the requirement for informed consent. If patients provided informed written consent to have data/samples from their medical records used in research, please include this information.'

4. Thank you for stating the following financial disclosure: "no"

5. Thank you for stating the following in your Competing Interests section:  "no"

Reviewers' comments:

Reviewer's Responses to Questions

**Comments to the Author**

1. Is the manuscript technically sound, and do the data support the conclusions?

Reviewer #1: Yes

Reviewer #2: Partly

2. Has the statistical analysis been performed appropriately and rigorously? 

Reviewer #1: Yes

Reviewer #2: Yes

3. Have the authors made all data underlying the findings in their manuscript fully available?

Reviewer #1: Yes

Reviewer #2: Yes

4. Is the manuscript presented in an intelligible fashion and written in standard English?

Reviewer #1: No

Reviewer #2: Yes

5. Review Comments to the Author

Reviewer #1: The study aims to compare four previously developed predictive scores for stone-free rate (SFR) after flexible ureterorenoscopy (f-URS) with holmium-YAG laser fragmentation of renal and ureteral lithiasis. The study was single-center and used between 669-800 procedures to evaluate the four predictive scores. Two scores have been previously validated- the Resolu Unsal Stone Score (RUSS) and modified Seoul National University Renal Stone Complexity (S-ReSC) score. The other two scores, the S.T.O.N.E score described by Molina et al. and Ito’s nomogram have not been validated in any other cohorts.

The study has a clear objective, the end point is concretely defined, and the analysis is straightforward. However, the generalizability of the findings is unclear especially since this is a single center study with fewer than 1000 cases and the use of a single rater. Overall, this manuscript sets the stage for future studies that can further validate the clinical utility of the predictive scores.

Major points:

1) Inclusion of the absolute numbers of patients would be informative for Tables 1 and 2. For example, how many patients are in UR and RO groups for Table 1? How many patients are in the Yes and No groups for Table 2?

2) It is very difficult for this reader to decipher information in Table 2. It is difficult to calculate the percentages by how the authors display the numbers. The authors describe the proportion of patients by no/yes and %yes. I believe it is easier to interpret the percentages if the authors present the fraction of patients who were yes/total patients. For example, the pre-stenting group is show as "207/385". It should be changed to 385/592 or 65% so that the reader can see more easily for how the percentages were calculated. This should be done for all of the fractions in Tables 2 and 3.

3) Table 3: STONE score section is very difficult to interpret. What do the numbers represent? I believe that it is unrealistic to expect the reader to be familiar with the STONE score section without an explanation for what the scores mean. Some context in the manuscript would be helpful so that the reader can understand the basics of the STONE score system.

4) Table 3 has error in reporting the data

Lower pole location- %yes = 85.7%, not 14.3% (yes group)

Lower pole location- %yes = 75.9%, not 24.1% (no group)

Stone number in different calyces > 1 - %yes = 22.8%, not 37.4% (yes group)

Stone number in different calyces > 1 - %yes = 37.4%, not 22.8% (no group)

Hydronephrosis- %absent = 61%, not 39% (yes group)

Hydronephrosis- %absent = 60.4%, not 39.6% (no group)

5) For a non-urology audience it is unclear what role a predictive score for SFR would play in clinical decision making. It would be helpful to address the following in the discussion: if there was a perfect predictive score that was 100% sensitive and specific, would a particular score keep a physician from intervening with f-URS or rather is the score meant to be helpful for patient counseling and managing patient expectations. Explicitly stating the clinical impact of using a predictive score would be useful context.

Reviewer #2: The authors evaluated four predictive scoring models for stone-free rate after flexible ureteroscopic urinary stone surgery. Major pitfall of the study is lack of systemic and consistent post-operative imaging to evaluate stone-free. I have some other concerns and questions.

1. Title: I think all of them are not nomograms, but they are predictive model or scoring system. Thus, the title should be revised with appropriate term.

2. Stone composition analysis is importance information when interpreting results of urinary stone surgery. Please, provide the results (at lease, proportion of uric acid stone)

3. How many surgeons involved the study? Didn't they have different patient selection or surgical outcomes? How did you adjust for surgeon effect?

Please, additionally provide brief description of their experience.

4. The major advantage of the study is concurrent comparison of 4 existing predictive models to predict stone-free after flexible ureteroscopic stone surgery. If you discuss it in practical aspect with balanced manner, not only focusing to the predictive ability, it can be very valuable discussion. For example, granularity of score or how many factors do they need can effect the predictive ability. By contrast, how difficult to calculate the score and how much time do they need can diminish the ability and clinical usefulness.

6. PLOS authors have the option to publish the peer review history of their article (what does this mean?). If published, this will include your full peer review and any attached files.

Reviewer #1: No

Reviewer #2: Yes: Chang Wook Jeong

---

## [Author Response · Author response to Decision Letter 0]

2 Jun 2020

Dear Dr. Joerg HEBER,

We thank the Editorial Board and the reviewers for their critical and empathic assessment of our work. Their comments allowed us to improve the quality of our work. In the following letter we address a point by point answer to their comments.

We hope that the Editorial Board as well as the expert reviewers will find our manuscript of interest for the readership of Plos One.

With my best regards,

Pr. François Kleinclauss, (MD, PhD) 

Point by point answer : All responses to corrector are specified in the file : "Responses to reviewers"

The file naming have been changed

2. In ethics statement in the manuscript and in the online submission form, please provide additional information about the patient records/samples used in your retrospective study. Specifically, please ensure that you have discussed whether all data/samples were fully anonymized before you accessed them and/or whether the IRB or ethics committee waived the requirement for informed consent. If patients provided informed written consent to have data/samples from their medical records used in research, please include this information.'

We Provide additional information in the online submission.

All data were anonymized

Ethics committee waived the requirement for informed consent.

The « Délégation a la Recherche Clinique et à l’innovation » approved the study. This institution is certified ISO9001v15.

We update this information.

ORCID iD for the corresponding author is : 0000-0002-1049-0117

4. Thank you for stating the following financial disclosure: "no"

 The authors received no specific funding for this work. The authors didn’t receive a salary.

The cover letter has been modified.

5. Thank you for stating the following in your Competing Interests section: "no"

 The cover letter has been modified.

“We have no conflicts of interest to disclose.” change to “The authors have declared that no competing interests exist.” in the cover letter.

Please know it is PLOS ONE policy for corresponding authors to declare, on behalf of all authors, all potential competing interests for the purposes of transparency. PLOS defines a competing interest as anything that interferes with, or could reasonably be perceived as interfering with, the full and objective presentation, peer review, editorial decision-making, or publication of research or non-research articles submitted to one of the journals. Competing interests can be financial or non-financial, professional, or personal. Competing interests can arise in relationship to an organization or another person. Please follow this link to our website for more details on competing interests:http://journals.plos.org/plosone/s/competing-interests

Reviewers' comments:

Reviewer's Responses to Questions

Reviewer #1: The study aims to compare four previously developed predictive scores for stone-free rate (SFR) after flexible ureterorenoscopy (f-URS) with holmium-YAG laser fragmentation of renal and ureteral lithiasis. The study was single-center and used between 669-800 procedures to evaluate the four predictive scores. Two scores have been previously validated- the Resolu Unsal Stone Score (RUSS) and modified Seoul National University Renal Stone Complexity (S-ReSC) score. The other two scores, the S.T.O.N.E score described by Molina et al. and Ito’s nomogram have not been validated in any other cohorts.

The study has a clear objective, the end point is concretely defined, and the analysis is straightforward. However, the generalizability of the findings is unclear especially since this is a single center study with fewer than 1000 cases and the use of a single rater. Overall, this manuscript sets the stage for future studies that can further validate the clinical utility of the predictive scores.

Major points:

1) Inclusion of the absolute numbers of patients would be informative for Tables 1 and 2. For example, how many patients are in UR and RO groups for Table 1? How many patients are in the Yes and No groups for Table 2?

We added the number of patients and the number of procedures in Table 1 : “Number of patients : 577 for UR group and 484 patients for RO group”

Nevertheless, each procedure was considered on an individual basis. 

In Table 2, we added the number of procedures in each group. 

2) It is very difficult for this reader to decipher information in Table 2. It is difficult to calculate the percentages by how the authors display the numbers. The authors describe the proportion of patients by no/yes and %yes. I believe it is easier to interpret the percentages if the authors present the fraction of patients who were yes/total patients. For example, the pre-stenting group is show as "207/385". It should be changed to 385/592 or 65% so that the reader can see more easily for how the percentages were calculated. This should be done for all of the fractions in Tables 2 and 3.

The suggested modifications have been made for Tables 2 and 3.

3) Table 3: STONE score section is very difficult to interpret. What do the numbers represent? I believe that it is unrealistic to expect the reader to be familiar with the STONE score section without an explanation for what the scores mean. Some context in the manuscript would be helpful so that the reader can understand the basics of the STONE score system.

The STONE score and the other scores are defined in the Material and Method section. The scoring system has been added in the table 3. We add line 103 “(5 to 15 points). A high total score is predictive of a failure f-URS outcome.”

4) Table 3 has error in reporting the data

Lower pole location- %yes = 85.7%, not 14.3% (yes group)

Lower pole location- %yes = 75.9%, not 24.1% (no group)

We checked all the data. After verification there was a little error in these data. However, these numbers correspond to Lower pole location associated with infendibulo-pelvic angle < 45°. 

71/482 (14,7%) of the procedures includes these parameters for ‘Yes stone free groupe’

43/187 (23.0%) of the procedures includes these parameters for ‘No stone free groupe’

Stone number in different calyces > 1 - %yes = 22.8%, not 37.4% (yes group) 

The suggested modifications have been made.

Stone number in different calyces > 1 - %yes = 37.4%, not 22.8% (no group) 

The suggested modifications have been made.

Hydronephrosis- %absent = 61%, not 39% (yes group)

Hydronephrosis- %absent = 60.4%, not 39.6% (no group)

The suggested modifications have been made. We reported the percentage of absent hydronephrosis because in this score, the absence of hydronephrosis is quoted.

Similarly, “lower pole calculi” change to “No lower pole calculi” in Ito’s score item, for easier interpretation.

5) For a non-urology audience it is unclear what role a predictive score for SFR would play in clinical decision making. It would be helpful to address the following in the discussion: if there was a perfect predictive score that was 100% sensitive and specific, would a particular score keep a physician from intervening with f-URS or rather is the score meant to be helpful for patient counseling and managing patient expectations. Explicitly stating the clinical impact of using a predictive score would be useful context.

We completely agree with this comment and add the following sentences in the discussion section line 254 : 

“Based on our results it’s difficult to rank this different scores, according there accuracy. We found that three factors i.e. stone size, stone number, lower pole and IPA < 45° mainly impact the scores results. Moreover, the remaining question is the clinical impact of such scores. Independently of the information gave to patients and maybe to help the choice of surgical technique this kind of scores are not clinically very usefull and are not routinely used for many reasons (difficulty and time to calculate). The RUSS score needs IPA angle definition on CT scan, the Ito’s score required the use of nomogram and the measurement of the stone volume, the STONE score can be calculated mentally in daily practice and the Seoul score would be the more easy to use.”

Answer to Reviewer #2 : 

The authors evaluated four predictive scoring models for stone-free rate after flexible ureteroscopic urinary stone surgery. Major pitfall of the study is lack of systemic and consistent post-operative imaging to evaluate stone-free. I have some other concerns and questions.

We agree with this comment and the absence of systematic post-operative imaging is one of the major bias of our study. This bias is reported in the discussion section. About a half of the patients underwent post-operative imaging but all underwent endoscopic and fluoroscopic inspection after surgery. 

1. Title: I think all of them are not nomograms, but they are predictive model or scoring system. Thus, the title should be revised with appropriate term.

We completely agree with this comment. 

The title was changed for “ Evaluation and comparison of scoring systems for predicting stone-free status after flexible ureteroscopy for renal and ureteral stones”

2. Stone composition analysis is importance information when interpreting results of urinary stone surgery. Please, provide the results (at lease, proportion of uric acid stone)

We add these data in the result section and reported the repartition of the stone composition. Line 146 : 

“Repartition of chemical composition : 66 % were calcium oxalate, 14% were calcium phosphate, 9% were uriq acid and 11% were other.”

3. How many surgeons involved the study? Didn't they have different patient selection or surgical outcomes? How did you adjust for surgeon effect?

Please, additionally provide brief description of their experience.

We add the number of surgeons line 237 : (12 operators)

These procedures were performed by residents, novices and experienced surgeons. This reflects real-life urological practice. The impact of experience and learning curve were not the goal in this paper but are the topics of another paper currently in writing. Moreover, we previouly reported that surgeon with less than 20 f-URS had similar results than surgeon with more than 20 f-URS procedures (Kleinclauss F, EAU2013) 

4. The major advantage of the study is concurrent comparison of 4 existing predictive models to predict stone-free after flexible ureteroscopic stone surgery. If you discuss it in practical aspect with balanced manner, not only focusing to the predictive ability, it can be very valuable discussion. For example, granularity of score or how many factors do they need can effect the predictive ability. By contrast, how difficult to calculate the score and how much time do they need can diminish the ability and clinical usefulness.

We agree with this comment and add a paragraph in the Discussion section about the routine use of such score (line 254) :

“Based on our results it’s difficult to rank this different scores, according there accuracy. We found that three factors i.e. stone size, stone number, lower pole and IPA < 45° mainly impact the scores results. Moreover, the remaining question is the clinical impact of such scores. Independently of the information gave to patients and maybe to help the choice of surgical technique this kind of scores are not clinically very usefull and are not routinely used for many reasons (difficulty and time to calculate). The RUSS score needs IPA angle definition on CT scan, the Ito’s score required the use of nomogram and the measurement of the stone volume, the STONE score can be calculated mentally in daily practice and the Seoul score would be the more easy to use.”

---

## [Decision Letter · Decision Letter 1]

26 Jun 2020

PONE-D-20-00720R1

Evaluation and comparison of scoring systems for predicting stone-free status after flexible ureteroscopy for renal and ureteral stones

PLOS ONE

Dear Dr. Kleinclauss,

Thank you for submitting your manuscript to PLOS ONE. After careful consideration, we feel that it has merit but does not fully meet PLOS ONE’s publication criteria as it currently stands. Therefore, we invite you to submit a revised version of the manuscript that addresses the points raised during the review process.

From a scientific point of view, this work is acceptable, however, it is suggested the improvement of the quality of the text as pointed out by Reviewer 1.

We look forward to receiving your revised manuscript.

Kind regards,

Antonio Riveiro Rodríguez, PhD

Academic Editor

PLOS ONE

Reviewers' comments:

Reviewer's Responses to Questions

**Comments to the Author**

1. If the authors have adequately addressed your comments raised in a previous round of review and you feel that this manuscript is now acceptable for publication, you may indicate that here to bypass the “Comments to the Author” section, enter your conflict of interest statement in the “Confidential to Editor” section, and submit your "Accept" recommendation.

Reviewer #1: All comments have been addressed

Reviewer #2: All comments have been addressed

2. Is the manuscript technically sound, and do the data support the conclusions?

Reviewer #1: Yes

Reviewer #2: Yes

3. Has the statistical analysis been performed appropriately and rigorously? 

Reviewer #1: Yes

Reviewer #2: Yes

4. Have the authors made all data underlying the findings in their manuscript fully available?

Reviewer #1: No

Reviewer #2: Yes

5. Is the manuscript presented in an intelligible fashion and written in standard English?

Reviewer #1: Yes

Reviewer #2: Yes

6. Review Comments to the Author

Reviewer #1: The following comments were adequately addressed:

1. The authors have added UR and RO patient numbers to Tables 1 and 2 and now provide helpful context.

2. The authors have provided fractions in Table 2 and now the data presentation is more clear.

3. The authors have provided a range for the S.C.OR.E system and they now provide an explanation for the meaning of a high score in line 103.

Minor Revisions:

1. We request that the authors take the time to make sure that the writing is clear and that grammar is used correctly. For example, there is a passage in the Discussion section that reads (starting line 259):

"Independently of the information gave to patients and maybe to help the choice of surgical technique this kind of scores are not clinically very usefull and are not routinely used for many reasons (difficulty and time to calculate)."

It is not clear to me what the authors are trying to say here. Please also spell check the document.

Starting line 261: "The RUSS score needs IPA angle definition on CT scan, the Ito’s score required the use of nomogram and the measurement of the stone volume, the STONE score can be calculated mentally in daily practice and the Seoul score would be the more easy to use."

There are 5 sentences here that are connected together into 1 large sentence. Please make this sentence more clear to the reader, perhaps the authors should break up the sentence into smaller sentences.

2. Please make uniform use of the decimal point or the comma. Please be consistent with the usage according to PLOS-ONE guidelines. For example:

Table 2, Lines 1 and 2 should read "25.9" and "27.9" as opposed to "25,9" and "27,9."

Table 3, missing a paranthesis before "14.7)" and should read "36.1" as opposed to "36,1".

3. In the Discussion section, there are three 1-sentence paragraphs and two 2-3 sentence paragraphs. In general, paragraphs should contain a collection of sentences that convey a related thought or message. The Discussion section could benefit from fewer paragraphs that are longer, rather than several 1-2 sentence paragraphs.

Reviewer #2: Thanks for the appropriate revision.

One minor thing should be corrected before publication; "uriq acid"  "uric acid".

7. PLOS authors have the option to publish the peer review history of their article (what does this mean?). If published, this will include your full peer review and any attached files.

Reviewer #1: No

Reviewer #2: No

---

## [Author Response · Author response to Decision Letter 1]

29 Jun 2020

Point by point answer :

Reviewers' comments:

6. Review Comments to the Author

Reviewer #1: The following comments were adequately addressed:

1. The authors have added UR and RO patient numbers to Tables 1 and 2 and now provide helpful context.

2. The authors have provided fractions in Table 2 and now the data presentation is more clear.

3. The authors have provided a range for the S.C.OR.E system and they now provide an explanation for the meaning of a high score in line 103.

Minor Revisions:

1. We request that the authors take the time to make sure that the writing is clear and that grammar is used correctly. For example, there is a passage in the Discussion section that reads (starting line 259):

"Independently of the information gave to patients and maybe to help the choice of surgical technique this kind of scores are not clinically very usefull and are not routinely used for many reasons (difficulty and time to calculate)."

It is not clear to me what the authors are trying to say here. Please also spell check the document.

The suggested modifications have been made. We change this passage to “ Based on our results it is difficult to rank the different scores according their accuracy. We found that four main factors: stone size, stone number, lower pole and IPA <45° affected the scores. Moreover, we should also ask what the clinical impact of these scores is. Independent of the information given to patients and possibly to help in the choice of surgical technique, these kind of scores are not clinically useful and are not used routinely due to their difficulty and the time required to calculate them.”

We also spell check the document.

Starting line 261: "The RUSS score needs IPA angle definition on CT scan, the Ito’s score required the use of nomogram and the measurement of the stone volume, the STONE score can be calculated mentally in daily practice and the Seoul score would be the more easy to use."

There are 5 sentences here that are connected together into 1 large sentence. Please make this sentence more clear to the reader, perhaps the authors should break up the sentence into smaller sentences.

The suggested modifications have been made. We change this sentence to : “The RUSS score needs IPA angle definition on a computed tomography scan and the Ito score requires the use of a nomogram and measurement of the stone volume. Conversely, the S.T.O.N.E. score can be calculated mentally in daily practice and the Seoul score would be easiest to use.”

2. Please make uniform use of the decimal point or the comma. Please be consistent with the usage according to PLOS-ONE guidelines. For example:

Table 2, Lines 1 and 2 should read "25.9" and "27.9" as opposed to "25,9" and "27,9."

The suggested modifications have been made.

Table 3, missing a paranthesis before "14.7)" and should read "36.1" as opposed to "36,1".

The suggested modifications have been made.

3. In the Discussion section, there are three 1-sentence paragraphs and two 2-3 sentence paragraphs. In general, paragraphs should contain a collection of sentences that convey a related thought or message. The Discussion section could benefit from fewer paragraphs that are longer, rather than several 1-2 sentence paragraphs.

The suggested modifications have been made. 

Some paragraphs have been grouped.

Reviewer #2: Thanks for the appropriate revision.

One minor thing should be corrected before publication; "uriq acid"  "uric acid".

This modification have been made.

The figure files have been upload to the Preflight Analysis and Conversion Engine (PACE) digital diagnostic tool.

---

## [Decision Letter · Decision Letter 2]

21 Jul 2020

Evaluation and comparison of scoring systems for predicting stone-free status after flexible ureteroscopy for renal and ureteral stones

PONE-D-20-00720R2

Dear Dr. Kleinclauss,

We’re pleased to inform you that your manuscript has been judged scientifically suitable for publication and will be formally accepted for publication once it meets all outstanding technical requirements.

Kind regards,

Antonio Riveiro Rodríguez, PhD

Academic Editor

PLOS ONE

Reviewers' comments:

Reviewer's Responses to Questions

**Comments to the Author**

1. If the authors have adequately addressed your comments raised in a previous round of review and you feel that this manuscript is now acceptable for publication, you may indicate that here to bypass the “Comments to the Author” section, enter your conflict of interest statement in the “Confidential to Editor” section, and submit your "Accept" recommendation.

Reviewer #1: All comments have been addressed

2. Is the manuscript technically sound, and do the data support the conclusions?

Reviewer #1: Yes

3. Has the statistical analysis been performed appropriately and rigorously? 

Reviewer #1: Yes

4. Have the authors made all data underlying the findings in their manuscript fully available?

Reviewer #1: Yes

5. Is the manuscript presented in an intelligible fashion and written in standard English?

Reviewer #1: Yes

6. Review Comments to the Author

Reviewer #1: All comments addressed from the previous review.

7. PLOS authors have the option to publish the peer review history of their article (what does this mean?). If published, this will include your full peer review and any attached files.

Reviewer #1: No

---

## [Editor Report · Acceptance letter]

28 Jul 2020

PONE-D-20-00720R2 

Evaluation and comparison of scoring systems for predicting stone-free status after flexible ureteroscopy for renal and ureteral stones 

Dear Dr. Kleinclauss:

I'm pleased to inform you that your manuscript has been deemed suitable for publication in PLOS ONE. Congratulations! Your manuscript is now with our production department. 

Kind regards, 

on behalf of

Dr. Antonio Riveiro Rodríguez 

Academic Editor

PLOS ONE